# Effects of Fear and Humor Appeals in Public Service Announcements (PSAs) on Intentions to Purchase Medications via Social Media

**DOI:** 10.3390/ijerph191912340

**Published:** 2022-09-28

**Authors:** Saleem Alhabash, Yao Dong, Charlotte Moureaud, Iago S. Muraro, John B. Hertig

**Affiliations:** 1Department of Advertising and Public Relations, College of Communication Arts, Michigan State University, East Lansing, MI 48824, USA; 2AbbVie, 1 N. Waukegan Road, North Chicago, IL 60064, USA; 3Department of Pharmacy Practice, College of Pharmacy and Health Sciences, Butler University, Indianapolis, IN 46208, USA

**Keywords:** substandard and falsified, prescription medication, social media, psychological reactance, risk perception

## Abstract

The increasing prevalence of online purchase of medications, specifically via social media platforms, poses significant health risks due to high chances of such medications being substandard and falsified (SF). The current study uses a 2 (persuasive appeal: fear vs. humor) x 3 (message repetition) mixed factorial experiment to investigate the effectiveness of persuasive appeals (on intentions to purchase medications online via social media referrals, mediated by psychological reactance (threat to freedom and anger), attitudes toward the public service announcements (PSAs), and viral behavioral intentions. ANOVA results showed the superiority of humor appeals compared to fear appeals in (1) reducing psychological reactance, (2) igniting favorable responses to the PSA, and (3) marginally reducing the intentions to purchase medications vial social media despite lower online engagement intentions (viral behavioral intentions). Pre-existing risk perceptions moderated these differences. A moderated serial mediation model, conducted using PROCESS models, was examined to assess the mechanism by which persuasive appeals and risk perceptions interact in influencing purchase intentions. Findings are discussed theoretically in regard to extending the psychological reactance model within the digital environment and practically in terms of public health, brand protection, and law enforcement recommendations.

## 1. Introduction

The online pharmacy market is expected to globally “cross revenues of over $141 billion by 2025, growing at a CAGR [Compound Annual Growth Rate] of over 20%” between 2020 and 2025 [1]. While several countries have legalized purchase of prescription medications from legitimate online pharmacies, a plethora of websites and dealers are supplying medications illegally, including substandard and falsified (SF) medical products. SF medical products deliberately or fraudulently misrepresent their identity, composition, or source [2]. With an estimated 2012 global trade of USD 431 billion [3], the threat of SF medications is exacerbated in low- and middle-income countries, where the WHO identified that “an estimated 1 in 10 medical products […] is substandard or falsified,” without accounting for online sales of medications [2].

The WHO estimated that 50% of all medications sold online were SF [4]. More troubling, the National Association of Boards of Pharmacy [5] identified over 1500 active sites as “Not Recommended,” most of which (over 94%) do not require a prescription and offer drugs illegally, thus directly contributing to the current opioid crisis. The COVID-19 pandemic accelerated the widespread reliance, worldwide and especially in the Unites States, on e-commerce to purchase medications. More than six in 10 U.S. consumers indicated likelihood and potential to purchase prescription and over-the-counter (OTC) medications from an online supplier [6].

Mimicking digital marketing strategies by pharmaceutical companies, illicit drug dealers and illegal pharmacies are leveraging social media to reach consumers [7]. Public policy recommendations and regulations have emphasized the vital role of brand protection specialists, law enforcement agencies, social media platforms, credit card companies, and delivery services to dismantle the public health crisis spurred by the online commercialization of SF medications [8].

Policy efforts should also emphasize consumer awareness and education initiatives [9]. Much like presenting health risk information in direct-to-consumer pharmaceutical advertising (DTCA) is vital to better inform consumers about the risks to their health and wellbeing [10,11,12], so is educating consumers about the dangers of purchasing medications online and via social media. This is especially true since research indicates an increasing prevalence of SF medications promoted on social media platforms like Twitter and Instagram [13]. Additionally, the prevalence of opportunistic online sales of SF medications coincides increase in deceptive advertising and marketing of SF medications as authentic and legitimate. This calls for scholarly and regulatory attention due to potential harm to on consumers’ health and wellbeing.

The current study contributes to the development of consumer-centric policy efforts by investigating the effects of public service announcements (PSAs) designed with fear and humor appeals on psychological reactance, attitudes, online, and offline behavioral intentions. This paper starts with reviewing the literature on medication sales via social media, including SF medical products. Next, we conceptualize fear and humor appeals. Third, we examine persuasive outcomes through the lens of psychological reactance and its effect on message evaluation and behavioral effectiveness. The fourth, fifth, and sixth sections describe the methods used in an online experiment, the results, and a discussion, respectively.

## 2. Literature Review

### 2.1. Online Medication Sales: The Case of Social Media

Within the context of Web 2.0 direct-to-consumer pharmaceutical advertising (eDTCA), Liang and Mackey [7] found that all top 10 global pharmaceutical companies had social media presence. Most of the highest-grossing drugs had brand-owned and non-corporate social media presence, thus indicating the prevalence of branded advertising of medications via social media. Moreover, illicit drug sellers use of such platforms to reach consumers more easily for SF medical products’ sales is central to the public health problem posed by the abuse of controlled substances and medications [8].

The use of social media and mobile messaging applications are increasingly used to access to medications [14]. A range of therapeutic drug classes are marketed without sufficient controls, including drugs for weight loss, attention deficit hyperactivity disorder (ADHD), steroids, inhalants, contraception drugs and devices, opioids, a variety of narcotics, and drugs in critical shortage [15]. Social media has, thus, become a hotbed for SF medications trading [16].

Users on social media platforms like Snapchat, Instagram, and Facebook use hashtags and emojis to access drug dealers both in terms of message promoting illegal drug/medication use as well as offering a supply channel for illegal sales [16]. Surveying drug users (international sample) that either used or considered using social media platforms to access drugs, Moyle and colleagues [14] found a wide range of platforms being employed to access illicit drugs. Snapchat and Instagram were the most commonly used platforms for illicit drug sales, which were motivated by convenience, accessibility, and ability to visually assess drug quality and transaction safety [14]. While the most purchased substances via social media were recreational drugs (e.g., marijuana, lysergic acid diethylamide [LSD], ecstasy/MDMA, mushrooms, and cocaine, respectively), drug users also reported purchasing prescription medications such as stimulants, opioids, and benzodiazepines, albeit at lower frequencies [14].

There is growing evidence of continued and increasing opioid marketing and sales on Twitter and Instagram without requiring prescriptions by different vendors, including illegal drug dealers [16]. Proactive data-based surveillance approaches (“infoveillance”) such as big data and machine learning have been studied as possible solutions to address this public health challenge. However, such efforts are hindered by lack of coordination and cooperation among multiple stakeholders [16]. Many infoveillance substance abuse studies have been conducted on Twitter, as it provides convenient access to data through its application programming interface (API) [17]. This explains the scarcity of studies on emerging popular platforms like Instagram, Snapchat, and TikTok. Using hashtag searches on Instagram, Li et al. identified 1228 posts (including pictures), where most hashtags and pictures referenced controlled substances like Xanax and oxycodone/OxyContin [17]. Such findings mirror survey findings. A quarter of a United Kingdom (UK) sample (16 to 24 years old) reported seeing advertisements for illicit drugs on social media, with the highest prevalence observed on Snapchat, Instagram, and Facebook [16]. Most ad exposures referenced recreational drugs (e.g., cannabis, cocaine, MDMA/ecstasy) and about one-fifth referenced prescription medications [16]. Four in 10 participants reported not being concerned about sales of medications over social media, with less concerns expressed by those under the age of 18 [18].

The sale of SF medications online and via social media platforms poses a significant economic and health threat that directly impacts individuals’ well-being. Amid growing complexity of the counterfeit supply chain, restrictions on effective takedowns, and increased difficulty of detecting counterfeit products, strategies to educate consumers (demand-side) are rising in significance and need [19,20]. A grounded-theory study of brand protection practitioners showed 12 different categories of tactics situated within the work of brand protection, and one of the most often-cited tactics was focused on communication to external (i.e., consumers) and internal audiences about counterfeiting [21].

Within the context of luxury counterfeit products, an exploratory study showed that a 30 min educational intervention focused on providing information about the counterfeit market enhanced participants’ efficacy in detecting counterfeits and influenced their attitudes and behavioral intentions regarding purchase of counterfeit luxury product [22]. In the pharmaceutical context, efforts focused on consumer education are on the rise. Chaudhry and Stumpf [23] provided cases of national and international organization’s efforts to educate consumers on how to detect counterfeit pharmaceuticals. Additionally, the authors report on a few pharmaceutical companies engages in consumer awareness campaigns via their social media accounts [23].

Attempts for anticounterfeiting consumer awareness and education, both in practice and in the academic literature are scarce, especially in relation to SF medications. Such interventions and campaigns are often heavy-handed on information and engage consumers in heightened state of threat appraisal by highlighting the dangers of buying SF medications. To date, and to the best of our knowledge, there has been no formal priori investigation of different persuasive appeals to guide brand protection specialists in their effort to educate consumers about the threats and dangers associated with buying SF, especially on social media. One study examined consumer responses—using in-depth interviews and surveys—to positive and negative counterfeiting awareness-raising messages and stratified messages by consumer types as they relate to counterfeiting. Such attempts, without sufficient and contextually relevant empirical evidence, might result in a boomerang effect. This is critical because a large body of health communication research emphasized cases when negative appeals, like fear, are less effective as they elicit negative emotional responses and could potentially lead to psychological reactance. The following section explicates fear and humor persuasive appeals.

### 2.2. Persuasive Appeals: Fear and Humor

Public service advertising deals with dissemination persuasive information related to a social problem with the intent to change people’s attitudes and/or behaviors [17]. PSAs leverage specific persuasive appeals, including logical, emotional, source-attribute, and fear, among others [24]. Emotional persuasive appeals, like fear and humor appeals are explicated as both a message feature and desired emotional response, where specific message attributes elicit desired emotional responses to facilitate message acceptance [25,26].

Fear is defined as an emotional state that is negatively valenced and high arousing that is activated when one faces a threat out of their own control [27,28,29]. Per Witte’s extended parallel process model (EPPM), a fear appeal message highlights the imminent threat resulting from noncompliance with the protective behavior in a way the evokes a negatively valanced response [30]. In contrast, humor “arises out of situations appraised as incongruous, unusual, or out of place” [29], p. 172. Skurka et al. explained that humor appeal persuasive messages are associated with garnering attention, increasing positive affect, and decreasing negative affect resultant from message experience [29].

To better understand the anticipated emotional response resulting from exposure to fear- and humor-appeal messages, the limited capacity model of motivated mediated message processing (LC4MP), based on Cacioppo and Gardner’s [31] bidimensional emotional processing model, provides a blueprint for how humans’ motivational system interacts with message features to produce cognitive and emotional responses [32]. Per the LC4MP, external cues (e.g., an emotional message) can activate the appetitive (approach) and/or aversive (withdraw) motivational system [32]. Based on findings from Lee and Lang, fear appeal messages are expected to produce negative emotions given aversive motivational activation, while humor appeal ones would produce a pleasant or positive emotional response given appetitive motivational activation [33]. Next, we review theoretical models adhering to each type of persuasive appeal.

The EPPM articulates that fear operates in parallel processes, depending on the interaction between persuasive arguments, perceived self and response efficacy, threat susceptibility, and severity. In cases when a message produces fearful emotional responses, humans resort to defensive motivational processing, thus leading to maladaptive changes (fear control). On the other hand, if a message does not elicit fear, then a parallel process activates protection motivation, thus leading to adaptive changes (danger control). Witte explicated that responding to a fear-inducing message is sensitive to perceived efficacy and threat susceptibility and severity [26]. An earlier model, protection motivation theory (PMT) is adapted into EPPM in which Rogers proposes three components of fear appeals: magnitude of noxiousness, probability of occurrence, and efficacy of recommended response, as well as self-efficacy; added later by Maddux & Rogers, whereas protection motivation arises from a cognitive appraisal of the three components in any given event [34,35]. Per the drive-reduction theoretical thinking, Williams argues that effectiveness of fear appeals depends on the balance between the seriousness of the threat and the message’s capacity to enable the receiver to resolve the fear by adhering to the adaptive, protective behavior (fear resolution) [36].

Findings are mixed regarding the effectiveness of fear-inducing messages. Often, the effectiveness of fear appeal depends on moderating variables that attenuate or magnify its impact on attitudes and behaviors. For example, Nabi, Roskos-Ewoldsen, and Dillman Carpentier found the effect of fear appeals was dependent on prior knowledge, where attitudes and intentions were higher for fear appeal messages when the level of prior knowledge was low, and the reverse was true for high knowledge participants [37]. In their meta-analysis, Peters, Ruiter, and Kok showed fear appeal and efficacy significantly interacted in affecting message-congruent behavioral outcomes, while noting mixed results and methodological flaws in this research stream [38]. A systematic review of fear-based appeals related to substance use showed that despite greater prevalence of studies documenting behavioral compliance with fear appeal messages, a notable portion of studies reported mixed and null effects [39]. Thompson, Barnett, and Pearce offer a qualitative view on the ineffectiveness of fear appeals that takes into consideration socioeconomic inequities that elevate feelings of stigma and shame resulting from exposure to fear appeal messages [40]. Emery and colleagues analyzed tweets voluntarily tagging the Centers for Disease Control and Prevention (CDC) “Tips from Former Smokers” campaign [41]. They found that the most relevant tweets reflected message acceptance, especially those high level of threat perception from the user [41].

While humor is widely prevalent in commercial advertising in North America, advertising and health communication scholars caution against its use without a clear strategy [42]. Sternthal and Craig argued that humor, in general, attracts attention, enhances message comprehension, distracts the audience, reduces counterarguing, and enhances source credibility and liking [43]. However, humor has weak effects on attitudinal and behavioral change [43]. More challenging for health communication message designers is portraying an issue of high severity in a light-hearted manner and tone, thus negatively influencing threat perception, and rendering a null effect on behavioral adherence.

Conway and Dubé found humor appeals to be superior to fear appeals in influencing behavioral adherence to condom use as a protective measure against HIV/AIDS [44]. Lammers et al. showed that when compared to serious appeals, humor appeals did not differ in evaluative outcomes immediately after ad exposure, yet upon delayed testing, were shown to be more effective than serious appeals [45]. Voss showed that use of humor appeals in advertising alongside with negative consequences was the most effective combination [46]. This message appeal combination significantly enhanced message comprehension, attitudes toward the ad, and net cognitions compared to ads that solely focused on negative consequences [46]. Blanc and Brigaud found that, compared to non-humorous ads for alcohol, tobacco, and obesity prevention, humorous ones enhanced attention to the message, memory, and ad persuasiveness [47]. Yoon examined the interplay between humor and shame appeals in human papillomavirus (HPV) vaccine ads [48]. Humor positively affected attitude and behavioral intentions specifically for individuals with lower levels of health worries [48]. Lewis et al.’s focus group participants favored positively framed ads to ones that were meant to induce fear and other negatively valanced emotions within the context of roadway safety ads/PSAs [49].

Brooker Jr. compared humor-, fear-, and information-based advertising and found that humor and information ads were similar, yet more favorable in evaluative outcomes to fear-based appeals in advertising [50]. Mukherjee and Dubé showed that adding a humor element to a fear appeal advertisement enhanced its persuasiveness [51]. Further, the authors found that the interaction between humor and fear was “mediated by defensive responses related to message elaboration and vulnerability to threat” [51], p. 147. Examining the 2007 French presidential elections, Capelli, Sabadie, and Trendel showed that humor appeals were effective in influencing attitudes toward the candidate among undecided voters [52]. Cacioppolo, Occa and Chunovskaya showed that colonoscopy screening humor messages were more effective than fear messages among participants with higher cancer worry [30]. Across different health contexts (e.g., alcohol abuse, smoking cessation), fear messages elicited greater interest and danger perceptions, while humorous messages were rated more favorably in terms of message and issue attitudes toward the message [53,54]. These differences were sensitive to individual differences like sensation. especially among low sensation seeking participants [53]. Shurka et al. found that both fear and humor appeals were equally effective in promoting climate change activism, yet the pathways to achieving message behavioral adherence was different [29]. Fear appeals affected behavioral intentions through higher activation of risk perceptions, humor appeals achieved this by influencing emotional responses [29].

The literature provides a mixed overview of the comparative effects of fear and humor appeals, but more importantly, such mixed findings on evaluative and behavioral outcomes are attributes to varying study contexts, accounting for both moderating variables (e.g., individual differences) and mediators. We anticipate a similarly complex view for this novel context subject of the current investigation: purchase of medications online. We argue that as consumers could vary in realizing the immediate risk to their own physical health, fear appeal messages might not be as successful as humorous ones. The basis of our prediction rests upon how fear inducing appeals affect psychological reactance and other persuasive outcomes.

### 2.3. Persuasive Outcomes and Hypotheses

Psychological Reactance Theory (PRT) articulates humans are motivationally aroused when their behavioral freedoms are threatened or eliminated, thus activating psychological reactance [55]. Reactance compels individuals to restore behavioral freedoms that were threatened or impounded. This motivational state commonly occurs when people perceive they are being told what to do through persuasive attempts.

Past research showed that exposure to messages the attempt to persuade individuals to refrain from risky behaviors, such as smoking, marijuana use, and consuming sugar-sweetened beverages activated psychological reactance, especially when the PSAs included a fear appeal [56,57,58,59,60]. Dillard, Kim, and Li found that the persuasiveness of anti-sugar-sweetened beverage PSA messages calling for policy change using fear appeal could have been improved by 17% if reactance was eliminated [60]. They also found that fear appeals had the potential to erode support for more policies restricting sugar-sweetened beverages [61].

Psychological reactance can enhance or inhibit message persuasiveness. In the current study, psychological reactance is a function of threat to freedom and negative emotions resulting from the message (anger). We postulate that fear appeal messages would result in greater threat to freedom and anger. Message cues are thought to activate or reduce psychological reactance, which mediates the relationship between message features and persuasive outcomes, such as motivation, attitudes, source appraisal, and ultimately behavioral intentions [54]. Based on this, we hypothesized:

**H1a.** 
*Participants exposed to fear appeal PSAs will express greater threat to freedom that those exposed to humor appeal PSAs.*


**H1b.** 
*Participants exposed to fear appeal PSAs will express greater anger than those exposed to humor appeal PSAs.*


In the current study, we focus on an expanded set of evaluative and behavioral persuasive outcomes. Per Alhabash, Mundel, and Hussain, persuasion models should be revised to fit the social media environment in ways that account for online engagement (termed as viral behavioral intentions [VBI]) that contribute to a linear information processing pathway in such a context [62]. With that in mind, in addition to examining psychological reactance, we also investigated the effectiveness of persuasive appeals on attitudes, VBI, and online medication purchase intentions.

As a concept, attitude toward the message is explicated as an individual’s tendency to respond favorably or unfavorably to a persuasive message [63,64]. Stemming from Janis’ drive model [65], and based on past research [54,65], we predict that participants will express higher favorability toward humor than fear appeal PSAs. Past research in health communication and advertising [45,46,47,48,49,50,51] showed that in both the short- and long-term, humor appeals yielded more favorable attitudes toward the message (e.g., PSA, ad) as well as the object of persuasion (e.g., brand, issue). Thus, we hypothesize:

**H1c.** 
*Participants exposed to fear appeal PSAs will express less favorable attitudes toward the PSA compared to those exposed to humor appeal PSAs.*


Based on Alhabash, Mundel, and Hussain [62], VBI refer to a message receiver’s intentions to engage with the message online through specific responses (e.g., like, share, and comment). Berge and Milkman conducted one of the earlier studies on information spread vial social media [66]. They showed that sharing of New York Times articles followed an emotional pattern of the content, where the more intense emotional content was shared more than milder forms of emotionality. Their investigation also showed that both positive and negative emotions were shared. Within the context of commercial advertising, it was found that positively valanced ads received greater VBI compared to negative ones [67]. However, recent investigations and with consideration of the evolving context of social media as well as regarding the specific context of this study, we anticipate that participants would express greater VBI for fear than humor appeal PSAs. This prediction is supported by big data analyses of social media content, where negatively valanced messages (e.g., tweets) were shared more rapidly and frequently compared to positive ones [68,69]. Finally, this study examines intentions to purchase medications online as a persuasive outcome, defined as “an individual’s conscious plan to make an effort to purchase a brand” or product [70], p. 55. Based on this, we hypothesize:

**H1d.** 
*Participants exposed to fear appeal PSAs will express greater VBI compared to those exposed to humor appeal PSAs.*


Past research has indicated that humor appeals are superior to fear appeals in terms of affecting behavioral adherence with the message’s persuasive arguments [29,44]. In the context of the current study, we argue that fear appeal PSAs, given the potential for activating reactance, might result in a boomerang effect, in that they would not be as successful as humor appeals in curbing participants intentions to purchase prescription medications via social media. Based on this, we hypothesize:

**H1e.** 
*Participants exposed to fear appeal PSAs will express greater intentions to purchase prescription medications via social media compared to those exposed to humor appeal PSAs.*


We also examined the moderating effect of perceived risks associated with purchase of SF medications via social media. Past research pointed to the effect of risk perception as an individual difference factor, on persuasiveness of risk and health communication messages [29]. As fear appeal messages elevate risk perceptions [29], we expect that risk perceptions measured prior to PSA exposure would interact with persuasive appeal in that those with lower risk perceptions to start with, will exhibit smaller differences between fear and humor appeals in relation to the five persuasive outcomes in this study. In contrast, those who had higher risk perceptions prior to PSA exposure will influenced more by fear than humor appeal PSA. In other words, we expect that lower risk perceptions will attenuate the effect of persuasive appeal, while higher risk perception will amplify those effect. Thus, we hypothesize:

**H2a.** 
*Participants with higher risk perceptions will express greater threat to freedom as a function of exposure to fear than humor appeals compared to those with lower risk perceptions.*


**H2b.** 
*Participants with higher risk perceptions will express greater anger as a function of exposure to fear than humor appeals compared to those with lower risk perceptions.*


**H2c.** 
*Participants with higher risk perceptions will express less favorable attitudes toward fear than humor appeals PSAs compared to those with lower risk perceptions.*


**H2d.** 
*Participants with higher risk perceptions will express greater VBI toward fear than humor appeals PSAs compared to those with lower risk perceptions.*


**H2e.** 
*Participants with higher risk perceptions will express greater intentions to purchase prescription medications from social media upon exposure to fear than humor appeals PSAs compared to those with lower risk perceptions.*


Finally, we tested a moderated serial mediation model for the effect of persuasive appeals and risk perceptions on intentions to purchase medications online through the serial mediation effect of threat to freedom, anger, attitudes toward the PSA, and viral behavioral intentions, respectively. We asked:

**RQ:** How, if at all, does risk perception moderate the effect of persuasive appeal on purchase intentions, mediated serially through psychological reactance (threat to freedom and anger), attitudes toward the PSA, and viral behavioral intentions, respectively?

## 3. Materials and Methods

### 3.1. Design and Participants

We used a 2 (persuasive appeal: fear versus humor) x 3 (message repetition) mixed factorial design with persuasive appeal manipulated between subjects and message repetition within subjects. This factorial design did not include a control group as the aim of the stdy was to compare the effectiveness of two persuasive appeals rather than examine the effects of whether or not a persuasive appeal was present. Per Mukerjee and Wu [71], the purpose of factorial design is to contrast two or more iterations (levels) of two or more factors/independent variables with the intent of dissecting main and interaction effects. This design is the most suitable to address the study’s hypotheses and research questions. Participants (*N* = 1002) were recruited through Amazon’s Mechanical Turk (AMT) between 19 August and 8 September 2020. Most participants identified as male (65.07%), followed by female (34.23%), with the remaining 0.69% identifying as transgender male/female, nonbinary-non-confirming, or other. The mean age for participants was 38.44 (*SD* = 11.91; *Range* = 21–100). About one third of the sample (34.13%) indicated they were of Hispanic or Latino origin. More than half of participants (53.99%) identified as White (including Middle Eastern/North African decent), followed by African American/Black (38.02%), Native American/American Indian/Alaska Native/Indigenous/Pacific Islander/Native Hawaiian (5.49%), Asian (4.39%), Latino-Hispanic (non-White; 3.69%), and Multiracial/Other (1.40%). Participants represented all 50 U.S. states, where most resided in urban (39.28%), suburban (26.45%), and rural (14.17%) areas. Most participants reported having at least a bachelor’s degree or above (88.52%), were married (80.74%), and currently employed (80.94%). The median income for the sample was USD 10,000 to 59,999 per annum. A total of 272 participants were discarded from the data set for unreliable answers. To assess the reliability of a participants’ responses, a team of four independent coders examined responses to seven different open-ended questions. A response was deemed unreliable if it appeared to have been lifted verbatim from an Internet source (e.g., definition of PSA), addressed an issue entirely irrelevant to the context of the study (e.g., a comment about prostate cancer), or nonsensical characters and/or words. Interrater reliability for the assessment of the responses’ reliability exceeded 90% agreement. to some of the open-ended questions, thus, the final sample size retained for analyses was 730 participants.

### 3.2. Independent and Moderator Variables

Participants were randomly assigned to view PSAs with either fear or humor appeals (persuasive appeal). To guard against message-specific effects, each persuasive appeal was represented with three message repetitions, presented at random (message repetition). Prior to stimuli (PSA) exposure, participants assessed their perceived risk of buying counterfeit medications by evaluating three statements using seven-point Likert-type scales that were developed for this study: (1) The risk when buying counterfeit medications is high; (2) Counterfeit medications are very dangerous; and (3) Purchasing counterfeit medications is risky. Factor (Eigenvalue = 2.13, % of Variance Explained = 70.96%, Factor Loadings = 0.833, 0.845, and 0.852, respectively) and reliability (Cronbach’s α = 0.795) analyses were satisfactory; the three items were averaged per participant. On average, participants ranged in their risk perception from 1.33 to 7 on a seven-point scale (*M* = 5.72, *SD* = 1.09).

### 3.3. Dependent Variables

All items were measured using a seven-point Likert-type scale, unless otherwise noted. Upon satisfactory factor and reliability analyses, items for each DV were averaged per message. For modelling results, means for all messages per appeal conditions were averaged into single DVs. Items and results of confirmatory factor and reliability analyses are reported in Table 1. We used MacKenzie, Lutz, and Belch’s three-item scale to measure attitude toward the PSA (A_PSA_) [64]. We used two scales from Dillard and Shen to measure psychological reactance: threat to freedom was measured with four items and anger was measured with four other items [72]. We used five items from Alhabash et al. to measure for viral behavioral intentions (VBI) [73]. Finally, purchase intentions (PI) were measured with four semantic differential scale items [70].

### 3.4. Control Variables

We controlled for gender identity (male/female/transgender male/transgender female/nonbinary or non-conforming/other), age (calculated from birth year), currently taking prescription medications, current health insurance status (yes/no), educational level, and household income.

### 3.5. Stimuli and Manipulation Check

We developed six PSAs (3 fear and 3 humor) using publicly available graphics (located through online searches) and messages developed by our research team. All stimuli were consistent in structure. Each stimulus included a visual with accompanying text made. Stimuli were designed to be of equal dimensions (600 × 800 pixels) using Adobe InDesign. The manipulation of fear and humor was executed using the images used in the PSA and the text accompanying the image (see Stimuli in Appendix A). Participants evaluated fear appeal messages as more scary (*M* = 4.97, *SD* = 1.52) than humor appeal ones (*M* = 4.04, *SD* = 1.89), *F*(1, 729) = 54.41, *p* < 0.001, partial-η^2^ = 0.07. Fear appeal messages were also rated as more negative (*M* = 5.07, *SD* = 1.46) than humor-appeal ones (*M* = 4.34, *SD* = 1.70), *F*(1, 729) = 38.39, *p* < 0.001, partial-η^2^ = 0.05 (appeal x repetitions interactions were not significant for both variables). The manipulation was deemed successful.

### 3.6. Procedure

The study was approved by the university’s Institutional Review Board (IRB). Participants were recruited through an AMT Human Intelligence Task (HIT). Data collection proceeded with a soft launch (N = 100) and a subsequent completion of the remainder of data collection (one week apart). Upon accepting the HIT, participants were directed to a Qualtrics online survey, where they were provided with an electronic version of the consent form and were also given the opportunity to download a digital copy of the consent form for their records. Afterwards, participants answered questions about their perceptions and behaviors related to purchasing medications online as well as other health-related questions. They were then randomly assigned to either the fear or humor appeal conditions, where they were exposed, at random, to three PSAs, and evaluated different DVs after each exposure. Finally, participants answered demographic questions (gender identity, birth year, race, ethnicity, education, employment, geographic location, and income). Upon completion of the study, participants were thanked and provided USD 1.51 for study completion.

### 3.7. Data Analysis

To test H1a-g related to the direct effects of persuasive appeals, we submitted the different DVs to 2 (persuasive appeal) x 3 (message repetition) repeated measures analysis of covariances (ANCOVAs), while accounting for control variables. As for H2a-g, we used a series of simple moderation models using Hayes’s PROCESS v3.5 macro for SPSS (Model 1) with 10,000 bootstrap samples [74]. Finally, to answer the research question, we used Model 92 [74] with persuasive appeal as an independent variable (IV), risk perception as a moderator, T2F, anger, A_PSA_, and VBI as mediators, and PI as a DV; a 100,000 bootstrap was used for this analysis as well.

## 4. Results

### 4.1. Descriptive Results

Most participants (91.53%) reported purchasing prescription medications at least once in the past 12 months, and the majority (77.60%) indicated that they have done so online. More than six in 10 participants (62.24%) indicated they are currently taking prescription medications, and about seven in 10 (69.77%) indicated they have discussed buying prescription medications online with a healthcare provider. The most popular platforms used to purchase prescription medications online were Amazon (54.43%), online pharmacy (44.61%), Facebook (32.33%), Instagram (26.33%), YouTube (19.92%), and WhatsApp (17.87%), among others (see Figure 1). For most of the platforms, participants majorly purchased narcotics (e.g., Vicodin, Percocet, Oxycontin, and Fentanyl) and stimulants (e.g., Adderall, Ritalin), followed by COVID-19 medications or vaccines and sedatives (e.g., Xanax, Valium, and Ativan) (see Figure 1).

### 4.2. Direct Effects of Persuasive Appeal

The main effect of persuasive appeals on A_PSA_ was significant, *F*(1, 721) = 16.40, *p* < 0.001, η^2^_p_ = 0.02, where participants expressed greater favorability toward the humor appeal PSA (*M* = 5.27, *SD* = 1.31) than the fear appeal PSA (*M* = 4.93, *SD* = 1.54). The following covariates had significant main effects on A_PSA_: gender, *F*(1, 721) = 6.20, *p* < 0.001, η^2^_p_ = 0.01, taking prescription medications, *F*(1, 721) = 13.42, *p* < 0.001, η^2^_p_ = 0.02, health insurance status, *F*(1, 721) = 6.62, *p* < 0.001, η^2^_p_ = 0.01, education, *F*(1, 721) = 36.54, *p* < 0.001, η^2^_p_ = 0.05, and income, *F*(1, 721) = 4.24, *p* < 0.001, η^2^_p_ = 0.01.

The main effect of persuasive appeal on T2F was significant, *F*(1, 723) = 14.97, *p* < 0.001, η^2^_p_ = 0.02. Participants expressed greater T2F upon exposure to fear appeal PSAs (*M* = 4.80, *SD* = 1.58) compared to humor appeal ones (*M* = 4.22, *SD* = 1.82). The following covariates had significant main effects on T2F: gender, *F*(1, 723) = 6.22, *p* = 0.01, η^2^_p_ = 0.01, taking prescription medications, *F*(1, 723) = 68.00, *p* < 0.001, η^2^_p_ = 0.09, education, *F*(1, 723) = 73.14, *p* < 0.001, η^2^_p_ = 0.09, and income, *F*(1, 723) = 12.48, *p* < 0.001, η^2^_p_ = 0.02.

The main effect of persuasive appeal on message-induced anger was significant, *F*(1, 723) = 7.87, *p* = 0.005, η^2^_p_ = 0.01. Participants expressed greater anger toward fear appeal PSAs (*M* = 4.46, *SD* = 1.86) compared to humor appeal ones (*M* = 3.95, *SD* = 1.91). The following covariates had significant main effects on anger: gender, *F*(1, 723) = 5.16, *p* = 0.02, η^2^_p_ = 0.01, taking prescription medications, *F*(1, 723) = 54.98, *p* < 0.001, η^2^_p_ = 0.07, education, *F*(1, 723) = 67.62, *p* < 0.001, η^2^_p_ = 0.09, and income, *F*(1, 723) = 14.62, *p* < 0.001, η^2^_p_ = 0.02.

The main effect of persuasive appeal on VBI was significant, *F*(1, 723) = 12.40, *p* < 0.001, η^2^_p_ = 0.02. Participants expressed greater VBI for fear appeal PSAs (*M* = 5.15, *SD* = 1.45) compared to humor appeal ones (*M* = 4.67, *SD* = 1.67). The following covariates had significant main effects on VBI: gender, *F*(1, 723) = 15.88, *p* < 0.001, η^2^_p_ = 0.02, taking prescription medications, *F*(1, 723) = 8.09, *p* < 0.001, η^2^_p_ = 0.01, education, *F*(1, 723) = 50.96, *p* < 0.001, η^2^_p_ = 0.07, and income, *F*(1, 723) = 10.55, *p* < 0.001, η^2^_p_ = 0.01.

The main effect of persuasive appeal on PI approached significance, *F*(1, 722) = 2.75, *p* = 0.098, η^2^_p_ = 0.004. Participants expressed greater intentions to purchase prescription medications through social media upon exposure to fear appeal PSAs (*M* = 4.60, *SD* = 1.99) compared to humor appeal ones (*M* = 4.22, *SD* = 2.04). The following covariates had significant main effects on PI: gender, *F*(1, 722) = 8.11, *p* = 0.005, η^2^_p_ = 0.01, taking prescription medications, *F*(1, 722) = 50.39, *p* < 0.001, η^2^_p_ = 0.07, education, *F*(1, 722) = 100.14, *p* < 0.001, η^2^_p_ = 0.12, and income, *F*(1, 722) = 16.21, *p* < 0.001, η^2^_p_ = 0.02. Means for all DVs by persuasive appeal are displayed in Figure 2.

### 4.3. Moderation Analysis

The second set of hypotheses focused on the moderating effect of pre-existing risk perception as it pertains to the effect of persuasive appeal on psychological reactance (threat to freedom and anger) (H2a-b), A_PSA_ (H2c), VBI (H2d), and PI (H2e). Each of the DVs was assessed with an individual, yet similar, simple moderation model with persuasive appeal as an IV, risk perception as a moderator, and the same control variables used in previous analyses. Additionally, the conditional effect of persuasive appeal on each of the DVs was assessed at various levels of risk perception (moderator) using the Johnson-Neyman significance regions (see Figure 3). The following report of the results is limited to assessing the interaction between persuasive appeal and risk perception (see Table 2 for full aggregate results of the five moderation models). Finally, interaction effects are plotted in Figure 4.

The interaction between persuasive appeal and risk perception was significant in affecting T2F, A_PSA_, and VBI, approached significance for anger, and was not significant in affecting PI. A similar pattern of findings (as shown in Figure 4) is observed for T2F and anger. The difference in T2F and anger between fear and humor appeal PSAs was attenuated at lower levels of risk perception. Yet, participants whose risk perception was high consistently reported greater T2F and anger upon exposure to fear appeal compared to humor appeal PSAs. It is worth mentioning that particularly for T2F, the Johnson-Neyman regions of significance also showed that for participants whose risk perception is extremely low (between 1.33 and 2.36 on the seven-point scale), they expressed greater T2F upon exposure to humor than fear appeal PSAs. With that in mind, H2a was partially supported and H2b was supported. In support of H2c, participants with low-risk perception did not differ significantly in their favorability of the PSA (A_PSA_), yet they expressed significantly more favorable A_PSA_ when the PSAs were designed with humor rather than fear appeal. As for VBI, similarly low-risk perception participants expressed similar VBI for fear and humor appeal PSAs, yet they expressed significantly higher VBI for fear than humor appeal PSAs when their risk perceptions were high; H2d was supported. The effect of the interaction between persuasive appeal and risk perception was not significant, hence, H2e was not supported.

### 4.4. Moderated Mediation Analysis

Results of the moderated serial mediation analysis are presented in Table 3 and Figure 5, set to answer the study’s research question. The final model explained 81% of the variance in intentions to purchase medications online. After controlling for the effect of moderators, mediators, and control variables, the direct effect of persuasive appeal on PI was not significant, while the direct effect of risk perception was significant. The direct effect of the interaction between persuasive appeal and risk perception was also not significant. VBI positively affected PI. Our findings showed that the addition of risk perception as a moderator altered the direction of the direct effect of persuasive appeal on T2F (humor > fear), A_PSA_ (fear > humor), and VBI (humor > fear). On its own, risk perception only directly affected PI, in that the greater participants’ risk perception, the lower their intentions to purchase medications online, expressed post-PSA exposure. However, risk perception successfully moderated the effect of persuasive appeal (as reported in the previous section) on T2F (negative), A_PSA_ (positive), and VBI (negative) (see Figure 4).

Moderator-mediator interaction between risk perception and threat to freedom positively affected PI, while the interaction between risk perception and VBI negatively predicted PI. The indirect effects analysis showed a trend in the mechanism with which PI is affected as a function of both persuasive appeal and risk perception, indicated by bootstrap regression coefficients at three levels of risk perception (lower, moderate, and higher). In most mediation and serial mediation cases, the indirect effects were significant only for the two higher risk perception participant groups, with the exception of the path from persuasive appeal through VBI to PI (negative indirect effect). T2F negatively mediated the moderated effect of persuasive appeal. On its own, persuasive appeal influenced T2F, A_PSA_, and VBI, respectively, which in turn positively influenced PI. A similar pattern was observed with the moderated mediation effect. The combinatorial effect of persuasive appeal and risk perception was manifested through a negative effect on T2F, which, in turn, positively influenced A_PSA_ and VBI, where VBI influenced PI. Additionally, both T2F and VBI significantly interacted with risk perception in influencing PI.

Finally, as shown in Table 3, indirect effects of the moderated mediation of persuasive appeal on PI was evident at higher values of risk perceptions (and not lower values) with regard to the mediational effect of T2F, A_PSA_, VBI, and serially with each of the following: T2F and anger; T2F and A_PSA_; T2F and VBI, T2F, anger, and VBI (respectively); T2F, A_PSA_, and VBI. With that in mind, anger is deemed an irrelevant response in the model’s contribution to PI.

## 5. Discussion

Our findings showed that humor-based appeals were more effective in reducing reactance (threat to freedom and anger), increasing favorability toward the PSA, and marginally decreasing intentions to purchase medications online. Fear appeal messages, on the other hand, increased intentions to share, like, and comment on the PSAs via social media (VBI). These differences between fear and humor appeals were mostly attenuated at low levels of risk perceptions associated with the purchase of counterfeit medications online. Yet, they were enhanced at higher levels of risk perception. Finally, the continuous linear ordering of different persuasive outcomes showed that the interaction between persuasive appeal and risk perception affected threat to freedom, which in turn affected VBI and purchase intentions, respectively.

Aside from the effect of persuasive appeal on VBI, humor appeals were more effective than fear appeals in reducing reactance and igniting favorable and message-congruent evaluative and behavioral responses to the PSAs. The case of VBI is of interest in this context. As noted earlier, the literature is mixed as to which type of content gets higher online engagement. Prosocial messages framed positively were found to result in greater online engagement intentions than negative ones [72]. However, studies examining the valence and emotionality of online content, specifically within the context of health information, have found greater prevalence and engagement of negative content [68,69]. At face value, our findings might be reflective of the lack of consistency as it relates to the valence of content with which users engage. However, it is also plausible that contextual and historical factors have influenced the pattern of findings presented in the current study. In this regard, the specific context of this study, framed as an illegal activity, is a largely negative context, thus adding humor to persuasive messages in this regard might not be as strong of a motivator for users to engage with this content online. On the other hand, it is also probable that the way social media have evolved recently, and the specific timeframe in which the study was conducted (pandemic and approaching presidential election in the U.S.), might also be reflective of ritualistic and habitual behavior among our participants (i.e., itis the norm to share negative content on social media). More interestingly, engaging with the PSAs online by expressing greater intentions to like, share, and comment on them, resulted in somewhat of a boomerang effect on the major behavioral outcome (purchase intentions), which runs counter to previous studies [75], where online engagement was positively associated with offline behavioral intentions. Given that fear appeal messages, which are reactance inducing to start with, were the ones receiving higher VBI ratings, it is important to research the motivations behind engaging with this type of content and how it influences behaviors.

Our findings related to the effect of risk perceptions is noteworthy. As shown in Figure 3, participants with lower levels of risk perception expressed the greatest levels of threat to freedom as a response to the PSAs. On the other hand, participants with higher risk perception levels expressed lower anger than their counterparts with low levels of risk perception. Additionally, those with higher levels of risk perceptions expressed more favorable attitudes toward the PSA and lower intentions to engage with the PSAs online (viral behavioral intentions). This pattern of findings mirror calls for a more tailored approach to persuasive health and risk communication based on individual differences congruent with the persuasive context. Past research, e.g., [76], showed that tailored and personalized risk communication enhances attitude change and behavioral compliance in public service and health communication campaigns. Past research [77] identified the prevalence of risk perceptions as a factor influencing counterfeit attitudes and behaviors, thus our findings support claims from this body of research.

Risk perceptions moderated the effects of persuasive appeals on four of the major DVs in the study. Participants with high risk perceptions expressed lower threat to freedom and anger for humor appeal messages. No differences were detected across different levels of risk perceptions when evaluating these constructs following exposure to fear appeal messages. As it relates to attitudes toward the PSA, risk perceptions enhanced favorable attitudes, where the higher the risk perception, the more favorable participants’ attitudes were toward the PSA. The reverse was true for viral behavioral intentions. Participants with higher levels of risk perceptions expressed greater VBI toward fear than humor appeal messages. Risk perceptions is not the only individual difference factor in the anticounterfeiting equation. Previous studies identified different consumer groups in relation to counterfeit behaviors (struggle, spurious, indifferent, and liberated consumers) [23]. Herstein and colleagues provided qualitative and quantitative evidence for the tailoring of messaging strategies as a function of individual differences and consumer groupings, where negative strategies resonated with struggle and spurious consumers, but not with indifferent and liberated consumers [23].

Finally, our moderated mediation analyses point to the potential of simultaneous persuasive processes involved with responses to PSAs promoting attitudinal and behavioral change for a risky issue. First, our findings showed that psychological reactance does not influence attitudinal and behavioral persuasive outcomes. However, persuasive appeals influence behavioral intentions (purchase intentions) through VBI. This effect was also successfully moderated by risk perceptions. This further enhances the significance of online engagement behaviors as a critical component in today’s persuasive processes. However, this finding should be interpreted with caution as it translates to awareness-raising campaigns related to curbing the prevalence and effects of SF medications. There is a potential of a boomerang effect, where higher VBI enticed participants to purchase SF medications through SF. This calls for a more targeted approach for future interventions by identifying specific key performance indicators of any campaign to enhance the effectiveness of such campaigns as they relate to risky behaviors.

### 5.1. Theoretical Implications

From a theoretical perspective, our findings support past research [30,53,54] that showed the importance of using humor appeals when communicating about risky behaviors with dire health impacts, like the ones associated with purchasing potentially SF medications from unverified and illegal online sources. Per past research [37], individual differences in relation to the object of persuasion (i.e., risk perception) successfully moderated the relationship between persuasive appeal and the different persuasive outcomes. Taken together, and in support of previous studies [60], the use of fear appeals in communicating about a risky behavior like purchasing medications online, can trickle a path of behavioral noncompliance due to the activation of psychological reactance, which in our study, was majorly facilitated by participants’ expression of threat to their freedom resulting from message exposure.

Another important theoretical insight from our findings deals with the importance of revising existing persuasion models to account for online engagement behaviors (e.g., VBI). Our findings showed that the effect of persuasive appeals on PI was mediated by VBI, thus but not by attitudes and psychological reactance, albeit that persuasive appeals did influence these outcome intermediary variables. Such a findings provides significant insights into a potential parallel process for understanding behavioral compliance, where, even with the presence of factors that hinder message compliance (e.g., psychological reactance), online engagement might activate pathways for behavior change. Alhabash, Mundel, and Hussain [62] have argued that in the age of digital transformation, it is critical to account for online behaviors in the persuasive reaction chain. Indeed, our findings reflected this, but showcased that while fear messages garnered greater VBI (higher intentions to like, share, and comment on the PSAs), such an effect negatively impacted adherence to the message’s argument, in that greater VBI were associated with greater purchase intentions, which is reflective of message noncompliance in our case.

### 5.2. Public Policy and Practical Implications and Recommendations

Anti-counterfeiting strategies to combat the illegal commercialization of SF medications have mostly focused on trade actors, including preventive measures, such as the adoption of authenticity seals for the identification of genuine products, and investigative efforts, such as international task forces to track and prosecute those involved in this criminal activity [9,78]. Despite many efforts among brand protection specialists, regulatory agencies, and law enforcement to curb the spread and prevalence of SF medications through proactive and responsive take-down strategies and concerted efforts to effectively break the supply chain for SF medications, it is critical to develop policies to purposefully engage consumers through strategic and tailored communication campaigns to raise their awareness and change their behaviors.

Traditionally, the policy focus on trade (versus consumer-centered) actions has been justified by pharmaceutical companies and regulatory agencies’ concern that unrestricted publicity about the problem could inadvertently undermine public trust in the pharmaceutical industry and the general effectiveness of medications [9]. Despite some legitimacy in this line of reasoning, our results indicate that well-thought-out public communication campaigns can contribute to the ongoing efforts to mitigate this public health problem and render public policies more effective. Our findings not only point to the importance of doing so but also to the complexity and need for evidence-based approaches to communicating with consumers.

From a communication strategy viewpoint, our recommendation to public policymakers and health practitioners concerning message appeal (fear versus humor) selection for PSAs is two-fold. First, this should be a data-driven decision with a priori assessment of the perceived level of risk. While both message appeal types examined in this study are likely to yield similar responses from target audiences with low perceived risk, different attitudinal and behavioral responses are expected among those with high perceived risk. Second, decision-makers should not lose sight of their primary desired outcome. While humor appeals might be slightly more effective in reducing purchase intention, fear appeals demonstrated relatively better VBI performance.

Last, if our moderated serial mediation model shows anything, it underscores the complexity of attitude and behavior changes. Therefore, residing to information provision only without accounting for message attributes and creative strategy, might render consumer education efforts a failure.

### 5.3. Limitations and Future Research

First, our findings, though experimental in nature, are not generalizable to a larger population due to our convenience sample. Future research should attempt to replicate our findings and recruit randomly selected samples to elevate the generalizability of these findings.

Second, we limited our investigation to fear and humor appeals, thus disregarding the potential effectiveness of other types of persuasive appeals (e.g., narrative, shame, guilt, etc.), which future research should emphasize.

Third, a significant portion of our sample were deemed to have provided unreliable responses (evidenced by open-ended question responses), while from our anecdotal communication with survey administration services, there is an increasing widespread of unreliable responses by both humans and bots, we do wish to emphasize that the portion of our sample that fell into this category could limit the validity our findings. However, we are confident that by removing these unreliable responses, our findings are satisfactorily reliable and valid.

Fourth, our experiment was conducted online. Though this experimental method enhanced ecological validity, given that the context of the study was social media, internal validity is limited given that the researchers were not able to control for all possible extraneous factors that could have influenced the findings. Participants completed the study online, which potentially entails potential errors, unknown to the researchers. For example, participants completed the study according to their own timelines, thus there is variability in the amount of time allocated for stimuli exposure and responding to DV questions. This could potentially harm the validity of our findings. However, the randomization to the two experimental groups (fear vs. humor appeals) and randomization of the order for the three message repetition levels enhances the study’s internal validity. Nonetheless, future research should replicate this study’s findings in a controlled environment. Fifth, our study mainly relied on self-report responses to DV measures, which could be sensitive to issues of social desirability and other threats to internal validity (e.g., historical error, regression o the mean error, etc.). Future research could incorporate unobtrusive measures of the psychological variables included in the study by leveraging eye-tracking and psychophysiological measures of cognitive and emotional responses.

Finally, given concerns from policymakers and stakeholders that public information about SF medications can undermine confidence in genuine drugs, future studies could extend our work by examining the effect of PSAs on public trust. It is also worth mentioning that we collected the data during the height of the COVID-19 pandemic, which was intentional on our part. However, the findings presented here could be context-specific; thus, there is a need for replicating our findings at a different time.

## 6. Conclusions

In the current study, we compared humor and fear appeals in anticounterfeiting PSA. Our study is one of few that focuses on consumer responses to educational and awareness-raising campaign attempting to curb the prevalence of counterfeiting via social media and change consumers’ attitudes and behaviors as it relates to purchase of counterfeit medications online and via social media platforms. Even though our findings point to greater effectiveness of humor appeals in terms of the persuasive outcome relevant to refraining from purchasing SF medications online, the mechanism with which such effects emerge are complex. In this study, we attempted to decipher part of this complexity by accounting for context-relevant individual differences—risk perceptions. Furthermore, our study pointed to the importance of online engagement (viral behavioral intentions) in driving intentions to purchase SF medications, which is worrisome and suggests a potential boomerang effect of awareness-raising campaigns in this domain. The scarcity of consumer-based research on counterfeiting calls for greater investment by the academic and professional communities in examining evidence-based strategies to influence consumers’ attitudes and behaviors.

## Figures and Tables

**Figure 1 ijerph-19-12340-f001:**
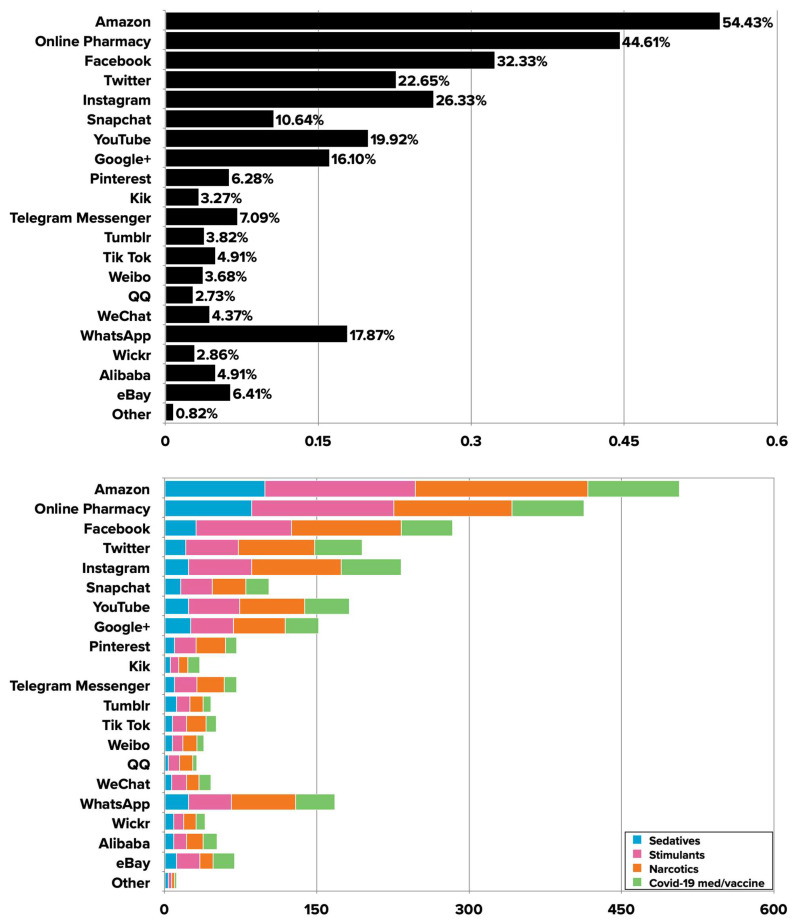
Percentage of participants who purchased medications on different platforms (**top**) and frequency of medication types purchased on different platforms (**bottom**).

**Figure 2 ijerph-19-12340-f002:**
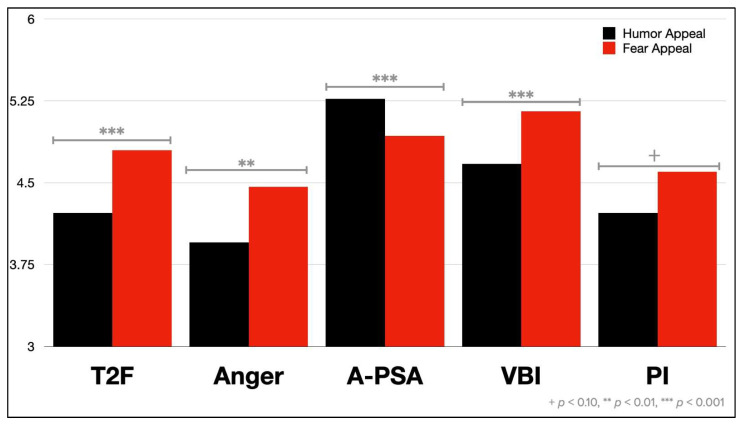
Means of major dependent variables, by persuasive appeal.

**Figure 3 ijerph-19-12340-f003:**
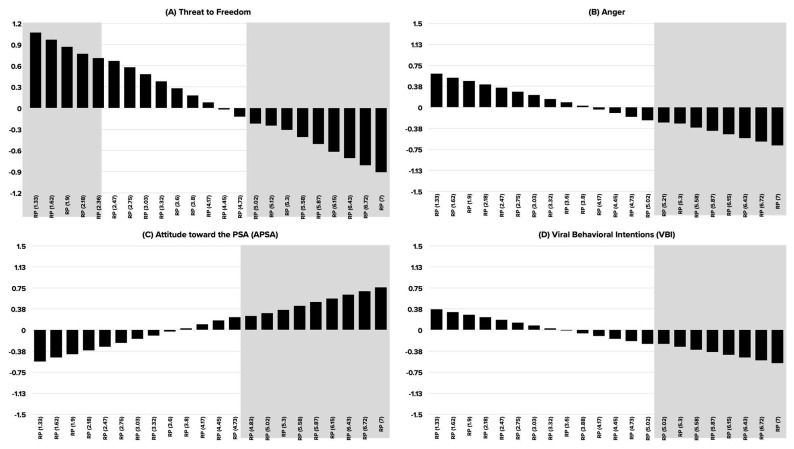
Johnson-Neyman regions of significance with unstandardized beta coefficients for the effect of persuasive appeal on DVs at different values of risk perception (moderator). Values reported in the graph indicate unstandardized beta coefficients. Shaded area indicates significant regions.

**Figure 4 ijerph-19-12340-f004:**
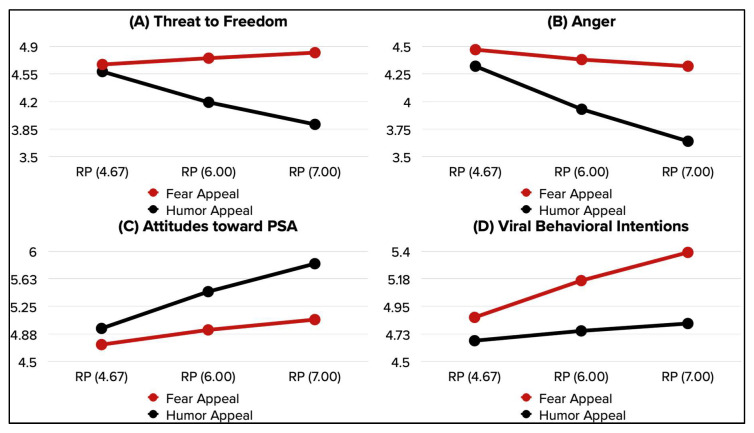
Moderating effect of risk perception on T2F, Anger, A_PSA_, and VBI.

**Figure 5 ijerph-19-12340-f005:**
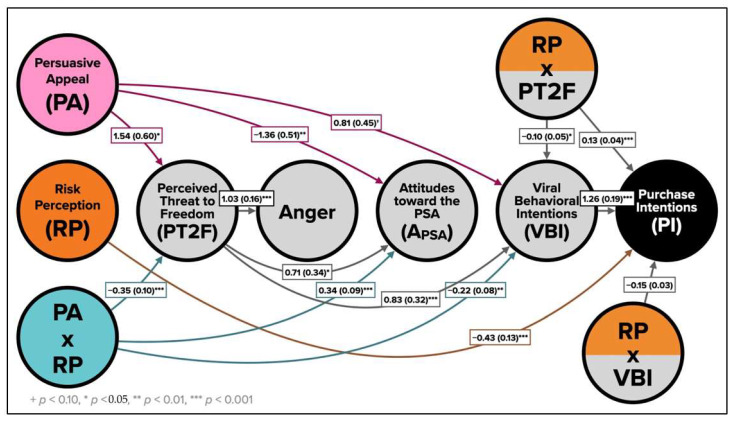
Moderated serial mediation of the effect of persuasive appeal on PI, moderated by risk perceptions.

**Table 1 ijerph-19-12340-t001:** Factor and reliability analysis results for all dependent variables.

	Statistic	Message 1	Message 2	Message 3
**Attitudes toward the PSA:** *This PSA is…* Bad/Good Negative/Positive Unfavorable/Favorable	Eigenvalue	2.56	2.62	2.56
	% of Var. Exp.	85.28%	87.22	85.54%
	Factor Loadings	0.919–0.928	0.932–0.935	0.923–0.928
	Cronbach’s α	0.914	0.927	0.915
	Mean (SD)	5.16 (1.59)	4.92 (1.74)	5.15 (1.59)
**Self-Efficacy** *After seeing this PSA, I am…* able to identify attempts for counterfeit medication sales on SM able to distinguish between counterfeit and legitimate medication on SM confident in my ability to identify counterfeit medications on SM	Eigenvalue	2.43	2.45	2.39
	% of Var. Exp.	80.94%	81.76%	79.63%
	Factor Loadings	0.892–0.909	0.901–0.908	0.884–0.899
	Cronbach’s α	0.882	0.888	0.872
	Mean (SD)	5.21 (1.40)	5.12 (1.45)	5.14 (1.40)
**Response Efficacy** *__________ would be an effective way to protect me from buying counterfeit medications* Simply avoiding to engage with medication sellers on SM By not clicking on a link to buy medications through social media Leaving a shady website selling medications	Eigenvalue	1.99	2.05	1.98
	% of Var. Exp.	66.20%	68.31%	65.86%
	Factor Loadings	0.759–0.855	0.769–0.855	0.726–0.858
	Cronbach’s α	0.739	0.762	0.731
	Mean (SD)	5.46 (1.15)	5.43 (1.19)	5.46 (1.15)
**Threat to Freedom** *This PSA…* Threatened my freedom to choose Tried to make a decision for me Tried to manipulate me Tried to pressure me	Eigenvalue	3.33	3.29	3.29
	% of Var. Exp.	83.36%	82.36%	82.33%
	Factor Loadings	0.907–0.921	0.899–0.916	0.882–0.914
	Cronbach’s α	0.933	0.928	0.928
	Mean (SD)	4.55 (1.80)	4.55 (1.78)	4.56 (1.79)
**Anger** *This PSA makes me feel…* Irritated Angry Annoyed Aggravated	Eigenvalue	3.57	3.52	3.59
	% of Var. Exp.	89.16%	87.94%	89.71%
	Factor Loadings	0.940–0.947	0.932–0.944	0.943–0.950
	Cronbach’s α	0.959	0.954	0.962
	Mean (SD)	4.22 (2.00)	4.29 (1.95)	4.23 (1.98)
**Viral Behavioral Intentions** This PSA is worth sharing with others I will recommend this PSA to others I will “like” this PSA on SM I will “share” this PSA on SM I will “comment” on this PSA on SM	Eigenvalue	3.97	3.97	3.96
	% of Var. Exp.	79.47%	79.30%	79.28%
	Factor Loadings	0.869–0.907	0.861–0.906	0.879–0.908
	Cronbach’s α	0.935	0.934	0.934
	Mean (SD)	4.94 (1.65)	4.92 (1.64)	5.00 (1.61)
**Behavioral Intentions** *How likely are you to buy medications through SM referral/link?* Not likely at all…Very likely Not probably…Very Probable Not certainly…Certainly Definitely not…Definitely	Eigenvalue	3.60	3.60	3.58
	% of Var. Exp.	90.05%	90.09%	89.45%
	Factor Loadings	0.941–0.958	0.942–0.957	0.936–0.955
	Cronbach’s α	0.963	0.963	0.961
	Mean (SD)	4.44 (2.07)	4.41 (2.06)	4.46 (2.07)

**Table 2 ijerph-19-12340-t002:** Simple moderation analysis results for the effect of persuasive appeal on all DVs, moderated by risk perception.

Predictor	T2F	Anger	A_PSA_	VBI	PI
	β (SE) [CI_LL−UL_]	β (SE) [CI_LL−UL_]	β (SE) [CI_LL−UL_]	β (SE) [CI_LL−UL_]	β (SE) [CI_LL−UL_]
constant	4.33 (0.62) *** [3.11, 5.54]	4.58 (0.70) *** [3.22, 5.95]	3.91 (0.56) *** [2.82, 5.00]	3.46 (0.60) *** [2.29, 4.63]	5.50 (0.72) *** [4.08, 6.93]
Persuasive Appeal (PA)	1.54 (0.60) * [0.36, 2.71]	0.90 (0.67) [−0.42, 2.23]	−0.87 (0.54) [−1.93, 0.19]	0.59 (0.58) [−0.54, 1.73]	0.50 (0.70) [−0.88, 1.88]
Risk Perception (RP)	0.06 (0.07) [−0.08, 0.20]	−0.07 (0.08) [−0.22, 0.09]	0.15 (0.06) * [0.02, 0.27]	0.23 (0.07) *** [0.10, 0.36]	−0.26 (0.08) *** [−0.42, −0.11]
PA x RP	−0.35 (0.10) *** [−0.55, −0.15]	−0.23 (0.12) † [−0.45, 0.002]	0.23 (0.09) * [0.05, 0.42]	−0.17 (0.10) † [−0.36, 0.03]	−0.14 (0.12) [−0.37, 0.10]
Gender	−0.23 (0.10)* [−0.43, −0.03]	−0.23 (0.11) * [−0.46, −0.01]	−0.27 (0.09) ** [−0.44, −0.09]	0.40 (0.10) *** [−0.59, −0.21]	−0.30 (0.12) * [−0.53, −.007]
Age	0.004 (0.005) [−0.01, 0.01]	0.01 (0.01) [−0.002, 0.02]	−0.001 (0.004) [−0.01, 0.01]	0.01 (0.005) [−0.004, 0.01]	0.002 (0.006) [−0.01, 0.01]
Prescription medication	−0.98 (0.12) *** [−1.22, −0.75]	−0.99 (0.13)*** [−1.25, −0.73]	−0.42 (0.11) *** [−0.63, −0.21]	−0.34 (0.12) ** [−0.57, −0.11]	0.99 (0.14) *** [−1.26, −.71)
Health insurance	−0.12 (0.13) [−0.37, 0.12]	−0.22 (0.14) [−0.49, 0.06]	0.23 (0.11) * [−0.45, −0.004]	−0.20 (0.12) [−0.43, 0.04]	−0.16 (0.15) [−0.45, 0.12]
Education	0.47 (0.06) *** [0.36, 0.59]	0.52 (0.07) *** [0.39, 0.64]	0.35 (0.05) *** [−0.25, 0.45]	0.40 (0.06) *** [0.30, 0.51]	0.66 (0.07) *** [0.53, 0.80]
HH income	−0.26 (0.07) *** [−0.40, −0.12]	−0.32 (0.08) *** [−0.48, −0.15]	−0.14 (0.07) * [−0.27, −0.01]	−0.24 (0.07) *** [−0.38, −0.10]	−0.34 (0.09) *** [−0.51, −0.18]
**Model Statistics**	R = 0.49, R^2^ = 0.24, *F*(9, 721) = 25.68 ***	R = 0.47, R^2^ = 0.22, *F*(9, 721) = 22.17 ***	R = 0.39, R^2^ = 0.15, *F*(9, 721) = 14.63 ***	R = 0.39, R^2^ = 0.15, *F*(9, 721) = 14.63 ***	R = 0.50, R^2^ = 25, *F*(9, 721) = 26.94 ***

Notes. DVS = Dependent Variables; PA = Persuasive Appeal; RP = Risk Perception; T2F = Threat to Freedom; A_PSA_ = Attitudes toward the PSA; VBI = Viral Behavioral Intentions; PI = Purchase Intentions; † *p* ≤ 0.10, * *p* ≤ 0.05, ** *p* ≤ 0.01, *** *p* ≤ 0.001.

**Table 3 ijerph-19-12340-t003:** Moderated serial mediation results for the effect of persuasive appeal on purchase intention, moderated by risk perception, and mediated serially by T2F, anger, A_psa_, and VBI.

Predictor	T2F β (SE) [CI_LL-UL_]	Anger β (SE) [CI_LL-UL_]	A_PSA_ β (SE) [CI_LL-UL_]	VBI β (SE) [CI_LL-UL_]	PI β (SE) [CI_LL-UL_]
constant	4.33 (0.62) *** [3.11, 5.54]	0.16 (0.78) [−1.38, 1.70]	2.07 (1.00) * [0.11, 4.04]	1.64 (1.00) [−0.32, 3.59]	0.78 (0.84) [−0.86, 2.42]
PA	1.54 (0.60) * [0.36, 2.71]	−0.49 (0.40) [−1.27, 0.29]	−1.36 (0.51) ** [−2.36, −0.36]	0.81 (0.45) † [−0.07, 1.70]	−0.13 (0.37) [−0.87, 0.60]
RP	0.06 (0.07) [−0.08, 0.20]	−0.05 (0.11) [−0.27, 0.17]	0.20 (0.14) [−0.08, 0.30]	−0.10 (0.15) [−0.39, 0.20]	−0.43 *** (0.13) [−0.68, −0.18]
PA x RP	−0.35 (0.10) *** [−0.55, −0.15]	0.09 (0.07) [−0.04, 0.23]	0.34 (0.09) *** [0.17, 0.52]	−0.22 (0.08) ** [−0.38, −0.07]	0.03 (0.07) [−0.10, 0.16]
T2F	--	1.03 (0.16) *** [0.71, 1.34]	0.71 (0.34) * [0.04, 1.38]	0.83 (0.32) ** [0.21, 1.45]	−0.41 (0.26) [−0.92, 0.10]
Anger	--	--	−0.30 (0.30) [−0.90, 0.30]	−0.38 (0.27) [−0.90, 0.14]	0.07 (0.22) [−0.35, 0.50]
A_PSA_	--	--	--	−0.01 (0.19) [−0.38, 0.37]	0.17 (0.18) [−0.18, 0.51]
VBI	--	--	--	--	1.26 (0.19) *** [0.88, 1.63]
T2F x RP	--	−0.02 (0.02) [−0.07, 0.03]	−0.06 (0.05) [−0.16, 0.05]	−0.10 (0.05) * [−0.20, −0.01]	0.13 *** (0.04) [0.05, 0.22]
Anger x RP	--	--	0.04 (0.05) [−0.05, 0.14]	0.08 (0.04) * [0.001, 0.17]	0.03 (0.03) [−0.04, 0.10]
A_PSA_ x RP	--	--	--	0.08 (0.03) ** [0.02, 0.14]	0.02 (0.02) [−0.04, 0.07]
VBI x RP	--	--	--	--	−0.15 *** (0.03) [−0.21, −0.10]
Gender	−0.23 (0.10)* [−0.43, −0.03]	−0.02 (0.07) [−0.15, 0.11]	−0.19 (0.09)* [−0.36, −0.02]	−0.18 (0.07) * [−0.33, −0.04]	0.02 (0.06) [−0.10, 0.14]
Age	0.004 (0.005) [−0.01, 0.01]	0.005 (0.003) [−0.001, 0.01]	−0.003 (0.004) [−0.01, 0.01]	0.003 (0.004) [−0.005, 0.01]	−0.003 (0.003) [−0.01, 0.002]
Prescription med.	−0.98 (0.12) *** [−1.22, −0.75]	−0.09 (0.08) [−0.25, 0.07]	−0.11 (0.11) [−0.31, 0.10]	0.20 (0.09) * [0.02, 0.38]	−0.09 (0.07) [−0.24, 0.05]
Health insurance	−0.12 (0.13) [−0.37, 0.12]	−0.10 (0.08) [−0.26, 0.07]	−0.18 (0.11) † [−0.39, 0.03]	−0.02 (0.09) [−0.21, 0.16]	0.03 (0.08) [−0.12, 0.18]
Education	0.47 (0.06) *** [0.36, 0.59]	0.08 *(0.04) * [0.004, 0.16]	0.20 (0.05) *** [0.10, 0.30]	0.07 (0.05) [−0.02, 0.15]	0.11 (0.04) [0.04, 0.18]
HH income	−0.26 (0.07) *** [−0.40, −0.12]	−0.08 (0.05) [−0.17, 0.02]	−0.06 (0.06) [−0.18, 0.06]	−0.08 (0.05) [−0.19, 0.03]	−0.03 (0.04) [−0.12, 0.06]
**Model Statistics**	R = 0.49, R^2^ = 0.24, *F*(9, 721) = 25.68 ***	R = 0.85, R^2^ = 0.73, *F*(11, 719) = 178.48 ***	R = 0.51, R^2^ = 0.26, *F*(13, 717) = 19.46 ***	R = 0.71, R^2^ = 0.51, *F*(15, 715) = 50.53 ***	R = 0.90, R^2^ = 0.81, *F*(17, 713) = 177.96 ***
**Indirect Effects ^a^**	**Risk Perception (Moderator) Values**				
	**Lower RP (4.67)**		**Moderate RP (6.00)**		**Higher RP (7.00)**
	**β (Boot SE)** **[Boot CI_LL-UL_]**		**β (Boot SE)** **[Boot CI_LL-UL_]**		**β (Boot SE)** **[Boot CI_LL-UL_]**
PA → T2F → PI	−0.02 (0.03) [−0.09, 0.04]		**−0.22 (0.06) [−0.35, −0.11]**		**−0.48 (0.13) [−0.74, −0.25]**
PA → Anger → PI	−0.01 (0.02) [−0.06, 0.02]		0.01 (0.02) [−0.03, 0.06]		0.04 (0.04) [−0.03, 0.13]
PA → A_PSA_ → PI	0.06 (0.03) [0.003, 0.14]		**0.18 (0.04) [0.11, 0.26]**		**0.28 (0.08) [0.14, 0.44]**
PA → VBI → PI	**−0.13 (0.06) [−0.25, −0.02]**		**−0.18 (0.04) [−0.28, −0.11]**		**−0.15 (0.05) [−0.26, −0.05]**
PA → T2F → Anger → PI	−0.02 (0.03) [−0.07, 0.03]		**−0.13 (0.04) [−0.21, −0.06]**		**−0.23 (0.09) [−0.43, −0.09]**
PA → T2F → A_PSA_ → PI	−0.01 (0.02) [−0.05, 0.02]		**−0.05 (0.−02) [−0.09, −0.02]**		**−0.07 (0.03) [−0.15, −0.02]**
PA → T2F → VBI → PI	−0.02 (0.03) [−0.09, 0.03]		**−0.04 (0.09, −0.001]**		−0.02 (0.02) [−0.06, 0.02]
PA → Anger → A_PSA_ → PI	0.002 (0.004) [−0.004, 0.01]		−0.001 (0.002) [−0.005, 0.003]		0.0002 (0.005) [−0.01, 0.01]
PA → Anger → VBI → PI	−0.001 (0.006) [−0.02, 0.01]		0.002 (0.004) [−0.01, 0.01]		0.006 (0.007) [−0.01, 0.02]
PA → T2F → Anger → A_PSA_ → PI	0.002 (0.005) [−0.01, 0.01]		0.01 (0.01) [−0.01, 0.02]		−0.001 (0.02) [−0.05, 0.04]
PA → T2F → Anger → VBI → PI	−0.001 (0.001) [−0.02, 0.02]		**−0.02 (0.01) [0.05, −0.001]**		**−0.03 (0.02) [−0.08, −0.004]**
PA → T2F → A_PSA_ → VBI → PI	−0.01 (0.01) [−0.03, 0.01]		**−0.03 (0.01) [−0.06, −0.01]**		**−0.03 (0.02) [−0.06, −0.01]**
PA → Anger → A_PSA_ → VBI → PI	0.001 (0.003) [−0.003, 0.001]		−0.0003 (0.001) [−0.003, 0.002]		0.0001 (0.002) [−0.003, 0.01]
PA → T2F → Anger → A_PSA_ → VBI → PI	0.002 (0.004) [−0.004, 0.01]		0.003 (0.01) [−0.01, 0.01]		−0.001 (0.01) [−0.02, 0.02]

Notes. PA = Persuasive Appeal; RP = Risk Perception; T2F = Threat to Freedom; A_PSA_ = Attitudes toward the PSA; VBI = Viral Behavioral Intentions; PI = Purchase Intentions; † *p* ≤ 0.10 * *p* ≤ 0.05, ** *p* ≤ 0.01, *** *p* ≤ 0.001; ^a^ bolded indirect effects indicate significant effect as confidence interval does not include a true zero.

## Data Availability

Not applicable.

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
