# Peer review of "Effects of Fear and Humor Appeals in Public Service Announcements (PSAs) on Intentions to Purchase Medications via Social Media"

_ijerph, 2022, doi:10.3390/ijerph191912340_

Round 1
Reviewer 1 Report (Previous Reviewer 2)
Thank you for the invitation to review the modified version of the paper.
The authors have made a number of consistent changes in their paper, but, from my point of view, a number of essential issues remain unresolved:
1. The authors did not break down their hypotheses formulated in the form of enumerations into distinct hypotheses, so that each of them is based on theoretical statements from the literature. In the case of quantitative empirical studies, the rigorous substantiation of the hypotheses is essential.
2. The research design is insufficiently explained. The authors claim to have used an experimental, factorial design. However, it does not appear whether a control group was used, (control missing) with which to compare the results obtained by exposure to the stimuli. This observation was also made in the initial feed-back, but in this resubmitted version remains unresolved. In the case of experimental designs, the entire procedure should be described in detail. This is an essential methodological problem of the study. Authors should include methodological references to support that the research design is appropriate.
3. In reporting the results of the research, the authors should also refer to the size effect. This should be explained so that the results are much better understood.
Author Response
I would like to thank the reviewer for acknowledging the revisions and improvement in the manuscript. Below, and on behalf of my co-authors, I would like to address the three comments raised in this round of revisions.
- The authors did not break down their hypotheses formulated in the form of enumerations into distinct hypotheses, so that each of them is based on theoretical statements from the literature. In the case of quantitative empirical studies, the rigorous substantiation of the hypotheses is essential.
RESPONSE: We have altered the presentation of hypotheses to accommodate the reviewer's comment.
2. The research design is insufficiently explained. The authors claim to have used an experimental, factorial design. However, it does not appear whether a control group was used, (control missing) with which to compare the results obtained by exposure to the stimuli. This observation was also made in the initial feed-back, but in this resubmitted version remains unresolved. In the case of experimental designs, the entire procedure should be described in detail. This is an essential methodological problem of the study. Authors should include methodological references to support that the research design is appropriate.
RESPONSE: As noted in our previous response to the reviewer's concern, a factorial design does not need a control group. We have included, in our previous response, a citation to support our argument. As noted earlier, a factorial design contrasts two levels of of two or more variables/factors. Given that our hypotheses are geared toward that particular contrast and how it is moderated by risk perception, a control group is not suitable to address the Hs and RQs. With that in mind, we have added a footnote in the Methods and Materials section where we describe the design.
3. In reporting the results of the research, the authors should also refer to the size effect. This should be explained so that the results are much better understood.
RESPONSE: We believe that we did include effects sizes in all our statistical analyses, where we indicated values for partial eta squared. If the authors believes there's a more suitable index of effect size, then please provide us with more specific guidance.
Round 2
Reviewer 1 Report (Previous Reviewer 2)
Thank you for the response. I consider that the authors answered to all my questions.
I hope that the comments helped them improve de quality of the paper.
Good luck!
This manuscript is a resubmission of an earlier submission. The following is a list of the peer review reports and author responses from that submission.
Round 1
Reviewer 1 Report
I have reviewed the paper carefully and you did good work. However, I recommend that you should do the following suggestions to improve the paper:
1. Some sentences were too long for readers and were not easily understand. You should reword these sentences, for example, line 63 to line 67, line 102 to line 105.
2. Data collecting instruments and data analysis methods should be added into the abstract.
3. Analysis results should not be written in the method section. The introductory features about the sample group are given in the method, but it may should be given in the findings section.
4. The discussion should be improved and supported according to the hypothesizes.
Reviewer 2 Report
Thank you for the invitation to review this article. Below you can find my comments and recommendations for the authors:
1. I would recommend that the authors better argue the research gap and present more sources from the literature that support the need to conduct studies in this area.
2. The research question should be placed before the research hypotheses.
3. Also, I would recommend that the research hypotheses be presented successively, based on the studies from the literature. The authors should consult more recent studies from the last 3 years, in order to better present the current state of the art.
Also, for example, H2: The effect of persuasive appeal on (a) threat to freedom, (b) anger, (c) attitudes toward the PSA, (d) viral behavioral intentions, and (e) purchase intentions – is not just one hypothesis, but 5. For a more rigorous approach, I would recommend that each hypothesis be written individually and argued based on the existing studies from the literature.
4. The authors should specify the source of the scale they used to measure the perceived risk of buying counterfeit medications and provide more information about the procedure used to develop the measurement model.
5. The authors should clearly present the control group they used.
They mention that ``the participants were then randomly assigned to either the fear or humor appeal conditions, where they were exposed, at random, to three PSAs, and evaluated different DVs after each exposure``. Therefore, these were the 2 experimental groups (Experimental group 1 and Experimental group 2). It is not clear what control group they used. Without a control group, the research design is not experimental, but rather quasi-experimental. The experiments result should compare the findings not only between groups but also with the control group. This is a serious methodological issue that must be clarified.
5. For Cronbach’s A is recommended to use the greek letter – α.
6. The actual theoretical contribution of this study is not clear. The authors should correlate the results of their research with the research gap. What is the originality of the research?
7. In the limitations section, the authors should also discuss the possible errors that can occur in the experimental design they used (e.g., historical error, regression to mean error, etc.). What procedure did they use to overcome these possible errors?
8. The conclusions of the paper should be extended.
Reviewer 3 Report
Overall Review: The research chosen and the application of the technique make the research interesting. Overall the research is well explained and presented and the work deserves appreciation. The research is very interesting and keeps holding the reader with the elaborations. This is very informative research with each paragraph. However, there may be some improvements in this work for the betterment of the study. The English language needs a minor improvement making sentences, and paragraphs shorter in the complete research paper, which may enhance the quality of the work. The work reflects the dedicated work of the research. However, suggested comments can help the researcher to improve the quality of the research paper for the mass reader.
(1) Title
· Authors have framed a very technical title, it is clear and accepted.
(2) Abstract and Keywords
· Abstract is well articulated which is representing a good snapshot of the research.
· Sentences are very well framed.
· Keywords are well chosen and suiting with the research.
(3) Introduction
· A good presentation of the work with the requirements.
· Line No. 34-36 must mention the country or world, and the geographic scope of CAGR.
· Sentences are written have a proper flow and connectivity is impressive.
· Research gap is well articulated.
· There are so many sections and subsections in the Introduction that reduce the quality of this research. It is recommended to modify this with the best possibilities.
· Some paragraphs are very large and must be smaller. Mostly the second paragraph of each section, 1.1.3. the first paragraph, and line no. 266-299.
· The middle part of the Introduction is giving the sense of a literature review. It may be modified.
· This section may be presented as two sections 1. Introduction, and 2. Literature Review or Background.
(4) Materials and Methods
· This segment is well explained with the clarity of explanations for sections and subsections.
· The research is using a good methodological approach.
· There is a need for citations.
· It is impressive and accepted as presented.
(5) Results
· The first paragraph has a detailed explanation.
· It is well explained with sub-sections displaying much clarity of result.
· Sub sections are too well organized.
· There is a need for citations.
· The study has enough statistical tests making the study more robust.
· The results are impressive with the presentation.
(6) Discussion
· Discussion is well presented with justifications.
· Two separate subheadings such as “Discussion”, and “Implications and Future Research” may be a better presentation.
· Line No. 551-575 needs to have more than 1 paragraph.
(7) Conclusions
· It is not enough as the conclusion, must have more elaboration.
· The presentation with bullet headings can make the conclusion more clear and attractive.
· It is better to present in several paragraphs to increase the credentials.